# The Outcome of Total Knee Arthroplasty for Patients with Psychiatric Disorders: A Single-Center Retrospective Study

**DOI:** 10.3390/medicina58091277

**Published:** 2022-09-14

**Authors:** Cristian Ioan Stoica, Georgiana Nedelea, Dragos C. Cotor, Mihai Gherghe, Dragos Eugen Georgescu, Christiana Dragosloveanu, Serban Dragosloveanu

**Affiliations:** 1Department of Orthopaedics, “Foisor” Orthopaedics Hospital, 030167 Bucharest, Romania; 2“Carol Davila” Faculty of Medicine, University of Medicine and Pharmacy, 050474 Bucharest, Romania; 3Department of General Surgery, “Dr. Ion Cantacuzino” Clinical Hospital, 022904 Bucharest, Romania

**Keywords:** TKA, osteoarthritis, psychiatry

## Abstract

*Background and Objectives*: For some years, psychiatric illness has been a major factor in evaluating the results of total knee arthroplasty. As with other patient-related items, patients diagnosed with mental illness have higher costs of medical treatment, longer recovery, and longer hospital stays. The aim of this paper is to evaluate the role of mental diseases on the surgical outcome compared with the normal population. *Materials and Methods*: At our hospital, we undertook a retrospective study between June 2020 and January 2022. The experimental group consisted of patients with mental diseases including schizophrenia, bipolar disease, depression, substance uses, or other psychiatric disorders. The control group consisted of patients who underwent total knee arthroplasty and did not have a mental disease. Postoperative complications and length of stay were also recorded during the study. We used the Western Ontario and McMaster Universities Osteoarthritis Index (WOMAC) and the Knee Society Score (KSS) as outcome measures. *Results*: Between June 2020 and January 2022, a total of 634 patients underwent total knee arthroplasty in our clinic, of which 239 had a mental disease. The majority of patients were female (61%), and the average length of stay was significantly longer for patients with mental illness (6.8 vs. 2.8 days). Preoperative WOMAC and KS function scores demonstrated statistically significant differences between groups (67.83 ± 17.8 vs. 62.75 ± 15.7 and 29.31 ± 19.8 vs. 34.98 ± 21.3). KS knee score did not show any significant differences preoperatively. All postoperative functional scores showed significantly better results for the control group compared to the mental illness group. *Conclusions*: Mental illness appears to be linked with lower TKA scores before and after the surgical procedure.

## 1. Introduction

Total knee arthroplasty helps to reduce pain, and is a standard surgical method to increase the quality of life in patients with severe osteoarthritis [1,2]. Patients report functional improvement and rapid recovery of mobility after total knee arthroplasty. Despite this fact, recovery in knee function is not the same for everyone [3]. In total knee arthroplasty, comorbidities and patient-related factors are more accurate prognostic factors than the chosen surgical technique and the type of implant used [4,5]. Postoperative results are usually linked with patient-related factors, some of which could lead to poor outcomes. Among these, we mention old age, high body mass index, female sex, as well as previous surgery and comorbidities such as high blood pressure, diabetes, kidney disease, and more [6,7,8]. For some years, psychiatric illness has been a major factor in evaluating the results of total knee arthroplasty. As with other patient-related items, patients diagnosed with mental illness have higher costs of medical treatment, longer recovery, and longer hospital stays [9,10]. Mental illness may also influence the preoperative time and the surgeon’s decision regarding the choice of implant for a better functional and psychological outcome [11]. In the past, psychiatric illnesses were underdiagnosed and associated with an increase in complications in total knee arthroplasty. Mental illness is accompanied by an increase in mortality and morbidity. In addition, preoperative depression is accompanied by low postoperative satisfaction and slower recovery [12]. A good selection of patients and a careful history of comorbidities are fundamental for a better postoperative outcome. However, many patients are underdiagnosed before total knee arthroplasty. Our hypothesis is that the knee function may be influenced by the patient’s psychiatric situation. The aim of this paper is to evaluate the role of mental diseases on the surgical outcome compared with the normal population.

## 2. Materials and Methods

At our hospital, we undertook a retrospective study between June 2020 and January 2022. Patients included in our paper were at least 60 years old. Multi-stage interventions and revisions were excluded. We also excluded patients who died before the last follow up or without any medical recordings available.

All patients who took part in this retrospective study underwent the same surgical procedure (medial subvastus approach), performed by the same surgeon. At the same time, they started physical therapy either at home or in a specialized clinic from the postoperative day.

The experimental group consisted of patients with mental diseases including schizophrenia, bipolar disease, depression, substance uses, or other psychiatric disorders. The control group consists of patients who underwent TKA and did not have a mental disease. Postoperative complications and length of stay were also recorded during the study. A total of 659 patients underwent total knee arthroplasty in our clinic, among which 25 were lost during the follow-up. The majority of patients were female (61%), and the average age did not show any statistically significant differences between groups (61.65 ± 8.34 vs. 62.35 ± 6.7).

### 2.1. Outcome Measures

Patients completed an outcome measure set preoperatively (with two weeks before the surgery) and 12 months postoperatively. We used the Western Ontario and McMaster Universities Osteoarthritis Index (WOMAC) [13] and the Knee Society Score (KSS) [14] as outcome measures. WOMAC is a questionnaire consisting of 3 subscales (pain, stiffness and physical function), which are also formed of 24 items. The WOMAC evaluates pain, stiffness, and function of the operated pelvic limb [15]. The KSS assesses pain, function, and range of motion of the knee. It is both a subjective and an objective outcome measure. The Visual Analog Scale (VAS) is a unidimensional measure of pain usually used to determine the pain level from a patient’s perspective. It also can be reliable in evaluating and correlating knee function and implant positioning [16].

The postsurgical physical examination is performed with great care in order to highlight signs of possible loosening of the prosthesis, signs of possible infection (swelling, tenderness), and, at the same time, to complete KSS.

For the radiological assessment, we used plain X-rays, including a lateral and an anteroposterior view. These were obtained preoperatively, and after 6 weeks, 3 months, and 6 months. We also obtained weight-bearing lower limb X-rays in order to develop preoperative planning. Evaluation of a possible component loosening is performed using the X-rays and the modified Knee Society radiology assessment [17,18].

### 2.2. Surgical Technique

The surgical technique was the same for all patients (medial subvastus approach), performed by the same surgeon. During surgery, we followed the international guidelines and the manufacturer’s implanting technique. Implantation was performed using a Zimmer Biomet NexGen LPS implant under a laminar flow system.

The type of anesthesia used was chosen between spinal anesthesia and general anesthesia depending on the patient’s disease history and age, as well as laboratory tests performed preoperatively. They were given antibiotics intravenously in the first twenty-four hours postoperatively as well as drug prophylaxis against deep vein thrombosis. All patients began rehabilitation from the first day after surgery.

### 2.3. Statistical Analysis

Continuous variables are presented in this paper as means and standard deviations (±SD). A *t*-test was performed in order to evaluate differences between preoperative and postoperative variables. In order to assess differences in categorical values, we used the Pearson chi-square test. We considered a *p* value lower than 0.05 as statistically significant. Statistical analyses were made with using SPSS version 27.0 and with the help of an independent statistician.

## 3. Results

Between June 2020 and January 2022, a total of 634 patients underwent total knee arthroplasty in our clinic, among which 239 had a mental disease. The majority of patients were female (61%), and the average age did not show any statistically significant differences between groups (61.65 ± 8.34 vs. 62.35 ± 6.7). The average length of stay was significantly longer for patients with mental illness (6.8 vs. 2.8 days). Baseline characteristics are presented in Table 1.

We broke down the mental health diagnoses in Table 2. Among 239 patients, 49 were diagnosed with bipolar disease, 96 with depression or anxiety, and 88 with multiple mental health disorders. Only 2 patients had been recorded with schizophrenia and 4 other patients presented other mental diseases. We did not find any patients to be addicted to substance use.

Functional results are presented in Table 3. Preoperative WOMAC and KS function scores demonstrated statistically significant differences between groups (67.83 ± 17.8 vs. 62.75 ± 15.7 and 29.31 ± 19.8 vs. 34.98 ± 21.3). Regarding preoperative KS knee scores, we recorded 39.78 ± 19.1 for the mental illness group and 42.36 ± 20.3 for the control group, but our study showed no statistically significant differences. All postoperative functional scores showed significantly better results for the control group compared to the mental illness group.

Regarding implant-related complications, we found 21 (8.78%) cases of periprosthetic infections among patients with mental illnesses, whereas the control group recorded 13 cases (3.29%) (*p* < 0.001). We recorded only 6 cases of periprosthetic fractures which required readmission. Among those cases, 4 belonged to the mental illness group.

## 4. Discussion

The results of this study confirm the presumption that patients with known preoperative mental illnesses are prone to medical and surgical complications. The World Health Organization predicts that by 2030, the main medical issue will be depressive disorders [19]. Apart from this issue, the number of total knee replacements per year is also increasing [20].

There are many studies which confirm that patients with preoperative psychiatric disorders are likely to have poorer functional outcomes after total knee arthroplasty [21,22].

As with other studies, our study shows a significant concordance between postoperative outcomes and preoperative mental illness. Numerous studies have shown that the length of stays in hospitals is longer for patients with mental illnesses [21,22]. Patients with mental illnesses tend to have a harder time adapting to recovery programs. They require a longer recovery time with a slower mobilization, greater dependence on family, and limited mobility [23,24]. Regarding these criteria, we recorded a significantly longer length of stay compared to the control group (<0.05). We found those results to be consistent with other findings in the literature [12,25,26].

Another clinically relevant contribution of our paper is that patients with mental diseases demonstrated a poorer WOMAC score, KS knee score, and function score compared to the control group before and after the surgical procedure. These findings are consistent with other papers that demonstrate the same results [22,27,28]. It also seems that patients with mental health issues consider themselves to have a worse clinical and functional status, thus pain perception could be linked to even more factors than just osteoarthritis changes [29,30,31].

Even though their results were poorer than the control group, the mental illness group showed statistically significant improvements. Some authors even noted that those functional improvements could have a beneficial effect on the mental health of patients, due to the fact that the level of pain could significantly lead to depression [30,32,33].

It is known that psychiatric diseases affect cognitive functions and increase the length of hospitalization, mortality, and the risk of serious complications. It has been found that delirium, after total knee arthroplasty, not only increases hospital length of stay and costs but also induces a decrease in function rehabilitation that leads to complications in the first postoperative year [34,35,36,37]. The very high costs of caring for patients with psychiatric illnesses can be explained by the large number of complications that occur after total knee arthroplasty [38]. In our study, we recorded a significantly higher risk of periprosthetic joint infections compared to the control group (*p* < 0.05) Among those cases, patients with severe psychiatric diseases are more at risk of such complications [4,7].

Six cases, among which four belonged to mental illness group, required readmission due to a periprosthetic fracture. This could also be related to underlying mental disease. Those findings are also supported by other papers available in the literature that found a correlation between injury susceptibility and underlying mental disease [39,40,41]. According to a study developed by Jørgensen et al. [22], psychiatric disorders are a risk factor for readmission caused by falls after TKA and THA. It is also considered a risk factor for postoperative anemia and pulmonary complications, which increase the length of stay [42].

Some limitations of this article must be mentioned when those results are being interpreted. First, the patients included do not represent the general population of our country. Those cases were surgically treated in just one medical center. Second, a longer patient assessment is required in order to evaluate functional results, and a larger cohort could be necessary to fully analyze the risk of local complications. In addition, this paper did not include medical postoperative complications, which may influence the postoperative results in some cases. Third, due to the fact that this is a retrospective study, there is only one data point before and after the surgery regarding the functional scores for each case. Thus, we were not able to fully track the progression. In the future, a large, multicenter prospective study could address all these limitations.

## 5. Conclusions

We have shown that patients with mental illnesses have longer hospital stays, increased complications, and more frequent readmissions compared to the control group. A careful assessment of mental status during the initial consultation is recommended to discover those patients who have a reduced mental state. We will recommend that patients with documented psychiatric illnesses who are undergoing total knee arthroplasty be referred to a psychiatrist first for relief of symptoms. These patients have an increased risk of being discharged much later than normal. In conclusion, mental illness appears to be linked with lower TKA scores before and after the surgical procedure.

## Figures and Tables

**Table 1 medicina-58-01277-t001:** Baseline characteristics.

	Patients with Mental Illness (n = 239)	Patients without Mental Illness (n = 395)	*p* Value
Gender (m/f)	102/137	145/250	n/a
Age	61.65 ± 8.34	62.35 ± 6.7	0.24
Left/Right	135/104	196/199	n/a
Length of stay	6.8	2.8	<0.05

**Table 2 medicina-58-01277-t002:** All mental health diagnoses recorded in our study.

Mental Health Diagnosis (n = 239) (%)	
Schizophrenia or other psychotic disorders	2 (1)
Bipolar disease	49 (20)
Depression	96 (40)
Substance use	0 (0)
Multiple health disorders	88 (37)
Other	4 (2)

**Table 3 medicina-58-01277-t003:** Preoperative and postoperative clinical scores.

Clinical Score	Preoperative			Postoperative		
	Mental Illness	without Mental Illness	*p* Value	Mental Illness	without Mental Illness	*p* Value
WOMAC score	67.83 ± 17.8	62.75 ± 15.7	<0.05	19.4 ± 25.3	8.3 ± 11.5	<0.05
KS knee score	39.78 ± 19.1	42.36 ± 20.3	0.1	78.34 ± 23.1	84.78 ± 21.1	<0.05
KS function score	29.31 ± 19.8	34.98 ± 21.3	<0.05	50.9 ± 22.3	57.8 ± 23.4	<0.05

## Data Availability

Not applicable.

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
