# Peer review of "The Outcome of Total Knee Arthroplasty for Patients with Psychiatric Disorders: A Single-Center Retrospective Study"

_medicina, 2022, doi:10.3390/medicina58091277_

Round 1
Reviewer 1 Report
Dear authors,
Although your research only focused on knee scores, it is valuable in terms of the number of subjects and mental patient groups. I believe that your research should be revised, especially in its statistical analysis. As for the other parts, I'd like to see your research again after all the edits I'll ask for some minor edits. The fixes I would like are as follows.
Title
Please write TKA in its long form, not an abbreviation.
Abstract
Please write the long version before the first TKA use and continue the abbreviation later.
Introduction
Line 32. Write as Total knee arthroplasty (TKA) and use only abbreviations for the rest of the text.
In the introduction, clearly demonstrate what previous research focuses on and what your research reveals differently.
Please give your hypotheses before the purpose statement.
Materials and Methods
-Please provide specific information about your subjects under the heading patients (how many subjects participated? How many completed the study? What was the average age, height, and weight of the experimental and control groups?), etc.
-Statistical Analyzes
I think the statistical analysis is wrong. Because in such retrospective studies, groups are not compared within themselves with pairwise comparison tests (the margin of error is very high). If you have two groups in pre-test and post-test research, mixed model repeated measurement gives the most meaningful results, and the margin of error is very low. In addition, some studies recommend the ANCOVA test. Please do the statistical analysis again. If not, prove scientifically why you didn't do it.
Discussion
I think your discussion is sufficient, but if there are different results after statistical analysis, revise the discussion section according to these results.
Best Regards
Author Response
Dear reviewer,
We thank you for all the fixes that you requested. We did all the modifications accordingly and attached to this answer.
Regarding the Statistical analysis section, we did not compare groups within themselves but we compared the differences between groups before and after surgery. A similar method was used by Lavernia et al in his paper called “What is the Role of Mental Health in Primary Total Knee Arthroplasty?” (Lavernia CJ, Villa JM, Iacobelli DA. What is the role of mental health in primary total knee arthroplasty? Clin Orthop Relat Res. 2015 Jan;473(1):159-63. doi: 10.1007/s11999-014-3769-5. PMID: 25002217; PMCID: PMC4390930.) They determined differences between the arthrofibrosis and control groups in the QWB-7, WOMAC, KS knee and function scores, active knee flexion, and KSROM before and after surgery using independent t-tests. Within the arthrofibrosis group, independent t-tests were used to determine if there were differences between psychologically distressed individuals and nondistressed individuals on PROs, knee scores, and objective motion measures. This is the reason why we did not use other methods.
Hope that this message is helpful. We deeply thank you for the suggestions and we are open to other suggestions or questions in order to improve our paper or to provide further clarifications.
Kind regards,
Dragos Cotor

Reviewer 2 Report
Dear Authors the topic is estremely interesting
As regards the introduction i suggest to improve this section describing the problem more deeper also in general for orthopedic prosthesis. In fact the psychiatric ilnesses influenced the otcomes but in preoperative time also the choice a part of surgeon. For this reason i suggest to cite the following article:
Plate vs reverse shoulder arthroplasty for proximal humeral fractures: The psychological health influence the choice of device?
Maccagnano G. et al.
World Journal of OrthopedicsOpen AccessVolume 13, Issue 3, Pages 297 - 30618 March 2022
As regards M&M, in “outocome measurements” i suggest to improve this sections speaking about the vas evaluation also. In fact it is important reaveals also the vas scale in order to understand if the knee prosthesis is painfull. Due to this aspect i suggest to cite the following article:
Painful knee prosthesis: CT scan to assess patellar angle and implant malrotation
Spinarelli A.
Muscles, Ligaments and Tendons JournalOpen AccessVolume 6, Issue 4, Pages 461 - 4661 October 2016
Furthemore i suggest to improve this section describing the complications that you detected
As regards the results and discussion, these sections are well described
Notwithstanding the limits of the study the conclusion is very important
Author Response
Dear reviewer.
We did all the revisions that you requested. Thank you for your kind feedback. Regarding the complications, they are mentioned in the results section. Thank you very much for the response. If you have any further sugestions or questions please send to us in order to increase the quality of our paper.
Kind regards,
Dragos Cotor

Round 2
Reviewer 1 Report
Dear Author,
Thank you for your efforts. Congratulations.